# Mental Workload Classification and Tasks Detection in Multitasking: Deep Learning Insights from EEG Study

**DOI:** 10.3390/brainsci14020149

**Published:** 2024-01-31

**Authors:** Miloš Pušica, Aneta Kartali, Luka Bojović, Ivan Gligorijević, Jelena Jovanović, Maria Chiara Leva, Bogdan Mijović

**Affiliations:** 1mBrainTrain LLC, 11000 Belgrade, Serbia; ivan@mbraintrain.com (I.G.); jelena.jovanovic@mbraintrain.com (J.J.); bogdan@mbraintrain.com (B.M.); 2School of Food Science and Environmental Health, Technological University Dublin, D07 H6K8 Dublin, Ireland; mariachiara.leva@tudublin.ie; 3Faculty of Computer and Information Science, University of Ljubljana, 1000 Ljubljana, Slovenia; ak83561@student.uni-lj.si; 4Microsoft Development Center Serbia, 11000 Belgrade, Serbia; lukajbojovic@gmail.com

**Keywords:** mental workload, mobile EEG, task load, multitasking, deep learning, pattern recognition, convolutional neural network, EEG signal classification, experiment

## Abstract

While the term task load (TL) refers to external task demands, the amount of work, or the number of tasks to be performed, mental workload (MWL) refers to the individual’s effort, mental capacity, or cognitive resources utilized while performing a task. MWL in multitasking scenarios is often closely linked with the quantity of tasks a person is handling within a given timeframe. In this study, we challenge this hypothesis from the perspective of electroencephalography (EEG) using a deep learning approach. We conducted an EEG experiment with 50 participants performing NASA Multi-Attribute Task Battery II (MATB-II) under 4 different task load levels. We designed a convolutional neural network (CNN) to help with two distinct classification tasks. In one setting, the CNN was used to classify EEG segments based on their task load level. In another setting, the same CNN architecture was trained again to detect the presence of individual MATB-II subtasks. Results show that, while the model successfully learns to detect whether a particular subtask is active in a given segment (i.e., to differentiate between different subtasks-related EEG patterns), it struggles to differentiate between the two highest levels of task load (i.e., to distinguish MWL-related EEG patterns). We speculate that the challenge comes from two factors: first, the experiment was designed in a way that these two highest levels differed only in the quantity of work within a given timeframe; and second, the participants’ effective adaptation to increased task demands, as evidenced by low error rates. Consequently, this indicates that under such conditions in multitasking, EEG may not reflect distinct enough patterns to differentiate higher levels of task load.

## 1. Introduction

Mental workload (MWL) is of considerable significance in various industries, particularly in high-stakes and safety-critical domains. In such settings, understanding and managing human mental resources are pivotal for ensuring an optimal performance and safety of operators. MWL assessment contributes to the better design of work environments and tasks that align with human cognitive capabilities. This, in turn, helps to prevent cognitive overload or underload, which can lead to errors, accidents, and reduced efficiency. In the fast-growing field of Brain–Computer Interface (BCI), an integrated, accurate, and reliable MWL assessment system brings great benefits. Such a system would enhance the efficiency and effectiveness of human–machine communication by adapting the system’s behavior based on the operator’s mental state. BCIs with real-time information about human MWL would allow dynamic adjustments to task complexity, system responsiveness, and feedback, making interactions more intuitive, seamless, and aligned with the operator’s cognitive capabilities.

MWL is a complex construct influenced by various interrelated factors, such as task load (task difficulty/complexity), environmental factors (noise, lighting, distractions, etc.), individual differences (cognitive abilities, training, expertise, etc.), physiological factors (fatigue, stress, etc.). However, in a given environment and limited timeframe, modifying individual and physiological factors is usually challenging. Hence, the primary determinants of MWL are the nature and demands of the task itself—referred to as task load (TL). A task often consists of a series of interconnected smaller subtasks, and the intricate relationship and dependency among these subtasks influence the resulting TL. The way these subtasks interrelate, their number, order in which they are executed and the complexity of each of the subtasks all play pivotal roles in shaping the overall perceived workload. It is intuitive and supported by research that increasing both the complexities of individual subtasks and the number of subtasks an operator manages within a given timeframe tends to elevate TL. However, the precise way in which these different factors influence TL remains unclear. The relationship between task quantity, task complexity, and resulting TL requires further investigation to uncover the specific mechanisms and dynamics at play.

### 1.1. Related Work

MWL assessment techniques can generally be categorized into three groups: subjective (questionaries, interviews, etc.), performance-based (accuracy, error rate, response time, etc.), and physiological measures (Electroencephalography (EEG), heart rate, galvanic skin response, etc.). However, when it comes to objective and real-time assessment, some studies point out EEG as the most relevant source of information [1,2,3]. It directly measures brain activity, offering real-time insights into cognitive processes. Furthermore, it is non-invasive, portable, and adaptable to various scenarios with high temporal resolution. The most widely used EEG analysis method for MWL assessment is the EEG frequency band analysis, which involves decomposing the brain’s electrical activity into different frequency components [4,5,6]. Each frequency band is associated with specific cognitive states, offering insights into cognitive processes and mental states based on the relative power within these frequency bands [7,8]. Particularly, specific EEG spectral components display consistent changes in reaction to cognitive demands of the task, showing a relationship between EEG spectral power and the task complexity. Namely, it has been observed that the frontal theta (4–7 Hz) power increases [9,10] and the parietal alpha (8–12 Hz) power [10] decreases as the task demands intensify. Based on these changes, another indicator has been derived—the ratio of frontal theta and parietal alpha power, a widely used EEG-based metric for MWL estimation [11,12]. This metric demonstrates correlation with objective TL measures across various range of tasks [13,14,15]. Other band powers like beta (13–30 Hz) [16], ratio of beta and alpha power [17], as well as various other power ratios across different bands are also used as indicators of MWL [18,19,20,21].

Research confirms that subjective evaluation of MWL correlates well with TL [19,22,23,24,25,26,27,28]. This is true when TL is modified through changes in subtasks complexities (qualitatively) [19,23,24,25], as well as through variations in the number of subtasks (quantitatively) [26,27,28]. However, the correlation between EEG frequency band-based metrics and TL varies depending on the nature of TL alterations. Specifically, these metrics demonstrate a notably stronger correlation with TL when TL is modified qualitatively wise as compared to quantitatively wise. It is important to note that, speaking of quantitative change, we assume that the task can be decomposed into smaller subtasks (like in a typical multitasking scenario). In that sense, a quantitative change of TL refers to increasing/decreasing the number of occurrences of subtasks, while a qualitative change of TL refers to modifying complexities of individual subtasks. In that sense, previous studies have reported significant correlation between the TL (which was qualitatively increased) and EEG frequency band-based metrics such as theta over alpha, where a positive correlation was reported [15], and alpha only where a significant negative correlation was shown [29]. The research conducted by Berka et al. [18] provided support for the effectiveness of EEG frequency band-based metrics as they used machine learning classifiers to define two indicators based on frequency bands recorded in different channels. Both indicators raised with escalating task demands. They independently tested 5 different tasks for several qualitatively altered TL levels, where the levels were qualitatively altered and found evidence of positive correlation for 4 out of the 5 tasks. On the other hand, several other studies [26,27,28] altered TL by changing number of subtasks in a giving time window using the NASA Multi-Attribute Task Battery (MATB) [30], tracking subjective MWL using the NASA-Task Load Index (NASA-TLX) questionnaire [31]. In those cases, the correlation between TL and subjective MWL was substantial while TL and the EEG frequency band-based metrics utilized by the authors did not exhibit any significant correlation.

Therefore, traditional EEG metrics for Mental Workload (MWL), based on simple frequency band analysis, often fall short of providing satisfactory outcomes in multitasking scenarios. This is particularly true when the number of occurrences of subtasks an operator handles is the predominant contributor of perceived MWL. However, the progression in machine learning [22], and particularly the rise of deep learning applied in the EEG field [32,33,34,35,36,37,38], is opening new possibilities for discovering relations between MWL and EEG signals. These approaches can bypass the intermediary steps of feature-engineering, leveraging the possibility of fully automatic end-to-end learning.

### 1.2. Research Overview

The objective of this paper was to harness the potential of end-to-end learning (with the help of a CNN) with the goal to examine which MWL-related EEG patterns could be differentiated in multitasking scenarios, particularly when task load (TL) varies quantitatively (based on the number of occurrences of subtasks within a given time frame). To achieve this, we proceeded with the following steps:(a)We designed an experiment with participants performing under several distinct levels of TL for the purpose of creating a dataset for EEG-based TL classification.(b)We employed the well-known and adaptable setting of the NASA Multi-Attribute Task Battery II (MATB-II, version 2.0) [39], which requires simultaneous management of multiple subtasks. Since the MATB-II software allows for easy customization of subtasks frequencies and distribution, the task was suitable for the design of the variable multitasking environment. Aside from controllability and trackability of the task, MATB-II is employed for its wide presence in the literature of the domain. It makes this study comparable to similar studies in the field. Furthermore, the experiment was designed to minimize participants’ physical load, ensuring that only their mental load is altered during the task. This also allows for the mitigation of EEG signal artifacts.(c)The experiment was designed in a way that changes the TL put upon an operator by changing the loads of individual MATB-II subtasks. Specifically, we designed 4 blocks of MATB-II subtasks combinations, representing 4 distinct TL levels by increasing/decreasing number of subtasks to be handled in a given timeframe. They were named Passive Watching (PW), Low Load (LL), Medium Load (ML), and Hard Load (HL) levels (reflecting increasing levels of difficulty). The first three blocks differed not only in task load, but also in the selection of active MATB-II subtasks. PW had no subtasks active, LL had 3 (out of 4) subtasks active, while ML and HL had all the subtasks active and differed only in the rate of occurrence of events to which participants were exposed. A description of the experimental design is reported in Section 2.3.(d)The EEG dataset was acquired against the predefined sequences of blocks representing the different TL levels, assuming that they would induce different MWL levels that could be detected by EEG. The environment was precisely controlled, with the activity of the MATB-II task software and the activity of the participants logged into separate text files synchronized with the collected EEG data for further analysis. These log files were vital for the data preparation for the model training (for the correct data labeling).(e)To distinguish between different TL levels, we employed a CNN. The input to the network was a short EEG segment and the output was a classified TL level (target class was the TL level assigned to the block to which the segment belonged).(f)The same model architecture was trained independently to detect the presence of each particular MATB-II subtask in a given segment. We wanted to test whether the same model was able to learn to distinguish between EEG patterns related to different subtasks’ activity. To the best of our knowledge, this is the first study to perform MATB-II subtasks detection from EEG. The results of this part of the study would also provide valuable insights for the application of EEG-based cognitive activity classification in the field of BCI, where distinguishing engagement in different tasks is relevant [40,41,42,43]. Additionally, good model performance in this part would further validate the model’s suitability for learning cognition-related EEG patterns, relevant to the problem of TL level classification The input to the model was also a short EEG segment and the output was a binary vector representing the activity status of each subtask.

Results for individual MATB-II subtasks detection were quite satisfactory—the detection accuracy of each subtask was reasonably high. On the other side, even though the results for TL levels classification showed good accuracy for PW and LL levels, the model struggled to distinguish between ML and HL levels. It is important to note that, although the quantitative TL was substantially different in these two levels, both levels had had all 4 MATB-II subtasks active. We speculate that this fact, together with the participants good adaptation to increased load during the whole experiment (as observed from low error rate), makes the distinction between the two levels challenging. This may indicate that EEG may not enclose the information about the increased TL in this kind of multitasking scenario.

## 2. Materials and Methods

### 2.1. MATB-II Task

Multi-Attribute Task Battery (MATB-II) is a computerized task designed to evaluate operator performance and workload. It encompasses a standardized set of tasks simulating activity performed by aircraft pilots. MATB-II requires concurrent execution of monitoring, tracking, listening and dynamic resource management tasks. Its multitasking paradigm aligns with the functionality of numerous real-world systems, making MATB-II a valuable tool for mental workload assessment across various domains [44].

The task includes 4 subtasks: System Monitoring (SYSM), Tracking (TRCK), Communication (COMM) and Resource Management (RMAN) (Figure 1a).

SYSM subtask requires monitoring of F5, F6 buttons, as well as F1, F2, F3, F4 scales and reacting if any anomaly occurs. Specifically, a participant is supposed to click on the corresponding button/scale if F5 (normally green) or F6 (normally red) change color, or if the yellow circle on the scales approaches the upper or lower edges of the bar. Frequency at which the buttons or scaler change their states can be programmed. Only a limited amount of time is given to respond to buttons/scales status change before it returns to normal by itself—if the participant fails to respond within that time, it is considered an error. The task is designed to mimic situations where operators are responsible for overseeing and controlling multiple aspects of a system or process.

In TRCK subtask, a participant is asked to use a joystick to keep the circle inside the boundaries of the smaller square in the center. The circle has some random drift and it is possible to program the speed of the drift. TRCK is assumed to replicate continuous compensatory actions inherent in piloting an aircraft.

COMM subtask simulates communication with air traffic control. A participant is supposed to listen to commands from audio output and act according to them. The command consists of information about which radio has to be changed and to what value the frequency has to be set. It should be noted that it is only necessary to respond to the command if the command starts with “NASA504”. Also, a limited time is given for the response. An error occurs if the time for the response passes or radios/frequencies are falsely adjusted.

RMAN is a more complex process control task that simulates abnormal situations or emergencies and allows for different strategies for problem solving [45]. The objective is to keep liquid levels of tanks A and B within desired boundaries by switching on/off 8 valves. The valves can sometimes break, in which case it is necessary to change a tank system strategy. It is considered an error if liquid levels in tanks A or B go out of the boundaries.

### 2.2. Experiment Setup

The experiment was conducted in a controlled setting, ensuring an optimal room temperature and the absence of potential distractions. Participants were seated in ergonomically designed office chairs, the position of which was adjustable relative to the desk (Figure 1b). The MATB-II task was presented on a touch screen monitor, strategically placed at an optimal viewing distance from the participant. Interaction with the task was facilitated using a joystick and touchscreen inputs. Before the experiment started, an EEG cap was carefully fitted onto the participant’s head. To guarantee the highest quality of EEG recording, electrode impedances were maintained below 10 kΩ.

### 2.3. Experimental Protocol and Task Design

Participants underwent a practice session (without data acquisition) a day before the EEG recording day. On the subsequent day, each participant engaged in the experiment, completing two sessions lasting 51 min. each, with a brief 10 min. rest between the sessions. A session consisted of a series of blocks comprising different combinations of the MATB-II subtasks, arranged so as to present different levels of task demand or task load (TL). There were 4 block types:Passive watching (PW): No activity was expected—the task was frozen and the participant would just look at the screen and wait for the next block;Low load (LL): All the MATB-II subtasks are active except TRCK, which was set to auto mode (no action required);Medium load (ML): All the MATB-II subtasks are active and the rates at which they would demand a response form the operator are increased compared to LL (see Table 1 for details);Hard load (HL). All the MATB-II subtasks are active and the rates at which they demand a response from the participant is about twice the ones in the ML block (except for COMM subtask). For instance, for RMAN in the HL block, the number of times valves turning on/off would be greater, as well as a higher liquid flow speed, requiring more attention from the participant. The exact number of occurrences of each subtask in a 5 min. block are given in Table 1.

There was no pause between the blocks, although the session would stop six times for a short period for the participant to complete NASA-TLX (see Figure 2). As there was no pause between the blocks, participants were also not informed when the task would go from one block to the next one. The PW block could last 1 min. or 2.5 min. The other blocks could last 5 min. or 2.5 min. The blocks were arranged in 4 different ways (configurations) to cover the 51 min. period of each session ensuring that in every session there were a total of 6 min. of PW blocks and 15 min. of each of LL, ML, and HL blocks (hence equivalent task load was faced in every session). The four different configurations of the blocks used in the sessions are displayed in Figure 2. During the experiment, all stimuli from the MATB-II program task, as well as participants’ inputs, with their respective times, were logged in text files, allowing for further data inspection, error detection, and analysis.

### 2.4. Participants

The study included 50 healthy participants, with normal or corrected-to-normal vision, within the age range of 18–39 years old (mean 25.9, std 5.4 years), comprising 21 males and 29 females, with exclusion criteria including any history of neurological disorders. None of the participants had prior experience with the MATB-II or BCI. All participants, regardless of their academic background, received detailed instructions about the experiment, in the form of a written manual. Following that, the instructor asked if everything was clear and answered any remaining questions. To confirm the participants’ understanding, a training session was conducted. Participants were selected using a convenience sampling method, primarily through networks of common acquaintances within the local community and university campus. All participants provided written consent before participating in the experiment and received monetary compensation for their involvement. The study was conducted in accordance with the Declaration of Helsinki and approved by the Ethics Committee of the Faculty of Medical Sciences, University of Kragujevac (protocol code 01-6471, 6 March 2021).

### 2.5. Equipment and Software

The EEG data was recorded using a wireless, 24-channel gel-based cap adhering to the standard 10–20 electrodes layout. The cap was connected to a mobile Smarting Mobi amplifier [46], produced by mBrainTrain LLC (Belgrade, Serbia) (https://www.mbraintrain.com, accessed on 10 November 2023). A sampling frequency of 500 Hz was employed for data acquisition. The reference electrode was set to FCz, and the ground electrode was placed at Fpz.

The task and participant’s activity and EEG data stream were synchronized with Streamer software (version 3.4.3, mBrainTrain) via the Lab Streaming Layer (LSL) [47].

### 2.6. Subjective MWL

After certain blocks, during a session, participants were given the NASA-TLX questionnaire to gather subjective workload assessments in various moments throughout the experiment. The NASA-TLX is a well-recognized evaluation tool for rating perceived workload. Participants used a visual-analogue interface, moving a slider to rate their task experience on an integer scale from low to high (from 1 to 100) across six variables: mental demand, physical demand, temporal demand, performance assessment, effort, and frustration. The results for variables relevant to our task are statistically analyzed to differentiate between the block types using ANOVA test and Tukey’s Honest Significant Difference (HSD) test.

### 2.7. EEG Pre-Processing

The objective in this research was to develop a deep learning model capable of learning directly from raw EEG data without relying on any hand-crafted features. Hence, only the basic pre-processing was done. No online filters were applied during the acquisition. While many EEG studies involve the removal of artifacts to improve signal quality and enhance the classification accuracy of deep learning models, as demonstrated in [48], our research intentionally omitted this step. Instead, we aimed to train an end-to-end model that learns directly from raw EEG data. The basic preprocessing steps are shown in Figure 3:

The details of each step are listed below:(1)Band-pass filtering (1–40 Hz)—to retain the frequency band related to brain activity;(2)Average re-referencing—to mitigate one-electrode-reference bias. The average of all the channels was added as the 25th channel to address the EEG data rank reduction issue [49] (reduced number of linearly independent channels due to re-referencing);(3)Channels standardization (by subtracting the mean channel value and diving by channel standard deviation)—to put the channels in the same scale and help better model training;(4)EEG signal downsampling from 500 Hz to 125 Hz, preserving the bandwidth of 40 Hz set by previous filtering.

Finally, these pre-processing steps set the stage for the deep learning model training, allowing the model to learn directly from the pre-processed raw EEG data.

### 2.8. CNN Architecture and Training Configuration

The model we used follows the representation learning paradigm, which means that it learns a representation of raw EEG data, reducing its dimensionality into vectors that contain essential information, while reducing redundancy of raw data. This representation is called an embedding and has a similar role as a vector of hand-crafted features used in traditional machine learning algorithms. Part of the network that learns embeddings from raw data is called the encoder network, while the part that learns target values (labels) from embeddings is called the decoder network. In practice, the decoder network is much simpler than the encoder network and it is so in our case, too.

The encoder we used is a convolutional neural network (CNN) with an architecture similar to the CNN encoder of the wav2vec model [50]. This design is motivated by the analogy between the audio signal (processed by wav2vec) and EEG signal, as both are inherently sequential in nature. A simple 2-layer fully connected (FC) network is used as the decoder. The best architecture hyperparameters are obtained using grid search. The grid search was performed across kernel sizes, stride lengths, number of layers, optimization algorithms, regularization techniques and types of loss functions. The model is visually presented in Figure 4.

Convolutions are performed along the time dimension. Dropout regularization and normalization are applied between layers and GELU/ReLU activation functions are used.

As explained previously, we used the same model architecture for TL level classification and for MATB-II subtasks detection. The only difference was in the last layer (last layer of the decoder)—its dimension, activation function, and loss function, depending on the type of the target we trained the model for. The details are given below:In the case of TL level estimation, we assigned each EEG segment to the block type it belongs to: PW, LL, ML, HL. Hence, the last layer dimension had 4 nodes and using a softmax function on top of it, we performed classification into the 4 classes. The loss function used was cross-entropy loss (Figure 4a).For MATB-II subtasks detection, the model output was a 3-dimensional bit-vector containing 1 or 0 in their respective positions if SYSM, COMM, RMAN subtasks individually were present/not present in the segment (the TRCK task was ignored for the detection as it is explained in Section 2.9.2). In accordance with that, the loss function was Binary Cross Entropy Loss (Figure 4b).

The implementation was carried out in PyTorch 1.10.1. The model underwent training for a duration of 35 epochs, based on observations of reaching optimal performance level within this epoch range. AdamW optimizer was utilized. A piecewise learning rate schedule was implemented: initially, the learning rate linearly increased over the first 5 epochs, reaching a value of 0.05. This was followed by a constant rate phase maintained up to the 20th epoch. Afterward, the learning rate decreased with a cosine curve pattern, eventually reaching a minimum of 0.005. The model required 48 min. of time per one training. The model of the machine utilized was BIZON G3000—Deep Learning Workstation (manufactured by BIZON, Hollywood, FL, USA) with the following technical specifications: 4 GPUs, processor 10-core 3.7 GHz Intel Core i9-10900X, memory DDR4 3000 MHz 128 GB (4 × 32 GB), 4 GPU-Ready (4 × NVIDIA Quadro; 1500W power supply), operating system BIZON Z-Stack, graphics card.

### 2.9. Dataset Labeling and Training Procedure

As previously explained, the model underwent training on two separate targets: TL level classification and MATB-II subtasks detection. This training was conducted independently for each target, meaning that CNN was trained from scratch separately for each one. The CNN was trained for TL level classification, and after obtaining results, it was reset and then trained again from the beginning for MATB-II subtasks detection (with slight modification in the architecture of the last layer shown in Figure 4). Experiment sessions, after only basic pre-processing described in Section 2.7, were divided into fixed-length EEG segments in the following way:TL level classification: 10 s segments, with 5 s overlapping. With 50 subjects completing two 51 min. sessions each, this resulted in the dataset of 61,100 segments;MATB-II subtasks detection: 15 s segments, with 10 s overlapping, resulting in the dataset of 61,000 segments.

For each target, the respective sets of segments were fed into the model as inputs, in two independent training stages.

#### 2.9.1. TL Level Classification Labeling

In this part, we detected TL level of a block a segment belongs to. Once EEG data was segmented, each segment was assigned a label according to the TL level of the block it is a part of. With four TL levels corresponding to 4 types of blocks, the segments were categorized into four classes: PW, LL, ML, and HL. For a clearer illustration, refer to Figure 2: all segments from the red sections would be labeled as HL, while those from the green, yellow, and white sections would be labeled as LL, ML, and PW, respectively. The segment length was 10s. This length was found as a trade-off between capturing sufficient MWL information within the segment and having big-enough number of segments for the model training. Longer segments hold more information about the average MWL of the block and hence are better for the estimation of the block TL level. However, using longer segments leaves us with fewer number of segments available for the model training. Furthermore, if we wanted to implement the estimation in real time, longer segments would introduce larger delay in the system.

#### 2.9.2. MATB-II Subtasks Detection Labeling

In this part, we detected a presence (activity) of a particular MATB-II subtask in a segment. It is important to remember that this detection and segments labeling was performed independently from TL level classification labeling, but solely based on which subtasks were active in a given segment. After the EEG data segmentation, each segment is labeled as a 3-dimensional bit-vector. It is assigned 1 or 0 in their respective positions in the vector as an indication of activity of each of the subtasks (SYSM, COMM, RMAN) in the segment. Subtasks presence in a segment is determined based on a participant activity, rather than the activity of a subtask in the MATB-II itself. For example, if SYSM was requiring an action in a given segment, but the participant did not take the action, we would assume that the task was overlooked, and would assign 0 for SYSM. The reasoning is legitimate since the SYSM action required but not taken as It was Ignored, would not have any impact on cognitive activity and EEG. Similar holds for COMM and RMAN. This way, we detect only active involvement in the subtasks. This type of classification task is known as multi-label binary classification.

SYMS is a discrete task, taking just a short notice and one click on the screen, meaning that it is obvious to differentiate its presence in a segment. However, COMM and RMAN tasks are continuous tasks and can be only partially in a segment. That is why we had to determine a threshold for the duration of these two subtasks within a segment to consider them active. Specifically, we assigned 1 for COMM or RMAN if the participants were engaged for at least 2 s. This interval was chosen as a result of grid search over different interval lengths, determining the best detection accuracy. It is a trade-off, as too short interval leads to assigning positive labels to segments not engaged with the subtask, while too long interval leads to assigning negative labels to segments engaged with the subtask. TRCK was not included in the detection task because it was not possible to discern the engagement in the subtask based on the logs available from MATB-II software. Namely, from TRCK logs, we only had the information about the distance of the circle center from the center of the screen (target square center) throughout time. However, we could not know whether the circle movement was caused by random drift or by the joystick (participant) activity and only the joystick activity implies active engagement with the subtask. Unfortunately, this information was not available from the MATB-II software. Nevertheless, the absence of TRCK from the detection task was not an issue for 3 other subtasks as the detection was independent.

Segment length (15 s) was chosen so that the dataset was balanced. This means that 45% of segments had 1 (positive) for SYSM, 38% had 1 for COMM and 52% had 1 for RMAN. The balance in the dataset was essential to prevent the model from being biased towards the majority class, ensuring better learning across both 1 and 0 classes.

#### 2.9.3. Training and Test Dataset Split

The same splitting procedure was applied to both TL level classification and MATB-II subtasks detection problems. After segmenting the sessions, all segments were split into training and test sets, ensuring that segments from any single session were exclusively in one set. This means that if a segment from a particular subject and session was allocated to the training set, then all segments from that subject and session were also in the training set. The same principle holds for the test set. Consequently, each session was assigned either to the training or the test set. Note that, as each subject completed two sessions, the number of sessions was twice the number of subjects. This approach prevented segments close to each other in time from being in the same set (training or test). This prevented data leakage and better reflected real-world scenarios. Subjects and sessions were split so that 5% of the data was used for test and the rest was used for training. Cross-validation was used to assess the model’s performance.

## 3. Results

### 3.1. Subjective MWL Assessment and Task Error Rate

The experiment was designed in a way that participants would experience varying levels of MWL related to different TL levels. In line with expectations, participants subjective MWL assessment through NASA-TLX showed higher MWL rates associated with higher TL blocks. Specifically, we examined 4 NASA-TLX variables relevant for the assessment of MWL: mental demand, temporal demand, effort, frustration, as well as their mean value. The goal was to analyze how participants subjectively assessed MWL in different block types. Since PW block type had no TL (participants were in resting state), this block type was excluded from the analysis, leaving us with 3 block types (classes) for examination: Low, Medium, and High. NASA-TLX questionaries were administered within every session in a way that there were two assessments of each of the three block types in each session (see Figure 2). In other words, in every session, for each of the three block types, two blocks of every type were assessed with NASA-TLX. Having 50 subjects completing two sessions each, this resulted in 200 assessments for each block type. The variables and their mean value were analyzed using one-way ANOVA (α = 0.05). After statistically significant difference among the 3 classes was confirmed for all variables, post hoc Tukey’s Honestly Significant Difference (HSD) test (α = 0.05) was employed to examine statistically significant difference in pairs of classes. The results of the analysis are shown in Figure 5.

ANOVA showed statistically significant differences among the 3 classes for each examined variable. Furthermore, Tuckey HSD test showed significant difference for each variable and for every pair of classes, except for medium vs. high comparison in effort variable. Therefore, since the participants were not told about the block type they were asked to fill out the questionnaire for, this confirms that they were able to recognize the task difficulty level.

Regarding participants performance on the task, analysis of the MATB-II logs showed that the task error rate was overall low (Figure 6). This means that participants managed to handle the task successfully across all TL levels.

### 3.2. TL Level Classification

TL level classification performance was estimated through the cross-validation method, using 5% of the data for model testing and the rest of the data for model training. Every 10s segment was labeled according to the block type it belongs to. So, we performed classification task into 4 classes: PW, LL, ML, and HL. Model architecture was identical to the model used for MATB-II tasks detection, with the only difference in the very last layer to accommodate the output size. The percentages of the classes in the dataset were 10%, 30%, 30%, 30% for PW, LL, ML, HL, respectively. Averaged classification results are presented with a confusion matric in Figure 7, together with the calculated F1 score and accuracy.

The confusion matrix gives us a good insight into the model performance, showing where the misclassified instances of each class were classified. Aside from the overall model accuracy, it is crucial to see which classes were mostly confused with each other. As expected, classes were mostly confused with the neighboring classes. The most accurately classified segments were from PW and LL classes. However, ML and HL classes had very low accuracy and were most predominantly confused with each other.

### 3.3. MATB-II Subtasks Detection

MATB-II subtasks detection performance was estimated also through cross-validation method, taking 5% of the data in test set and the rest for the training set. For every 15 s segment, we estimated the presence of each of SYSM, COMM and RMAN subtasks. Subtasks distribution was such that 45% of segments had SYSM, 38% had COMM and 52% had RMAN (of course, some segments had none of these tasks, some had one, two, or all three subtasks). The training set was balanced, but not perfectly balanced. For that reason, we presented the results using F1 score along with precision, recall and accuracy, to provide a complete overview. The averaged results for are shown in Table 2, while the distribution of results across subjects is presented with box plots in Figure 8.

Receiver operating characteristics (ROC) curves for the detection of the subtasks altogether and each subtask individually are presented in Figure 9.

It is important to note that some subtasks (COMM, RMAN) were considered present in the segment even if the segment only partially covered the subtask activity. That made the detection more challenging since the model had only a part of the subtask-related EEG pattern available to decide about the presence of the subtask (see Section 2.9.2). Good detection results for the individual subtasks indicate that the model was able to learn the overall cognitive activity-related EEG patterns of the segments. This implies that the encoder network successfully mapped raw EEG signals into lower dimension embeddings, that the decoder could further process to do the detection task.

## 4. Discussion

### 4.1. Subjective MWL Assessment and Task Error Rate

MATB-II subtasks are straightforward, primarily demanding participants’ timely action, rather than deep problem solving. The task load within the blocks was modified by adjusting the occurrences of subtasks activations within a given timeframe. An error is typically an indication that the participant has not paid timely attention to the subtask. It could be caused by the engagement with other subtasks or simply due to the lack of attention. Analysis of the MATB-II software logs showed that the participants were making very few errors in general. This means that they successfully managed TL in all the block types. Various studies in the literature [26,27,28,51] showed that subjectively experienced MWL estimated through self-assessment questionaries tends to increase with the increasing quantitative TL. This is also verified with our results, showing statistically significant relation between NASA-TLX scores and block task loads.

### 4.2. TL Level Classification

Results regarding the TL level classification are presented in the form of a confusion matrix. This provides a deeper understanding of the model performance, showing which classes are being confused with one another (falsely classified). Not surprisingly, classes are confused mostly with the neighboring classes (that have the most similar TL level). The most apparent observation from the confusion matrix is low accuracy for ML and HL blocks with a significant level of confusion between them. The literature recognizes similar problem when trying to differentiate between higher levels of MWL in multitasking, facing lower accuracies compared to differentiation of lower levels using machine learning approach [52,53] as well as traditional EEG MWL frequency band metrics [27,28,54].

Moreover, even though the TL and subjective NASA-TLX MWL scores were different in ML and HL, the model struggled to distinguish between them. One could argue that the inconsistency in frequencies of subtasks’ occurrences at a shorter timescale might lead to confusion between the classes. However, it’s important to note that MWL is not solely dependent on the number of actions in a given moment. The screen monitoring focus required by MATB-II is a significant factor of MWL, too. The task was designed in a way that higher TL block (HL) necessitate a quicker monitoring switch from subtask to subtask (compared to lower TL block (ML)), thereby contributing to higher MWL. This reasoning justifies the experiment hypothesis that higher TL blocks should induce higher MWL levels. On the other hand, low error rate shows that the participants successfully managed varying levels of TL in the sessions. Additionally, satisfactory model performance tested on MATB-II subtasks detection verifies the model’s capability to learn complex cognition-related EEG patterns. Considering all these aspects, it implies that, in the case when participants effectively adapt to quantitatively varying TL levels, EEG patterns do not reflect the changes in those levels. It would be interesting to conduct the study with the modified experiment, where additional overload block type would be added, with the TL level high enough that the participants would start making more errors. That would introduce a novelty in the experiment as the block type would reflect a situation where participant would be unable to handle the TL. Addressing the same TL level classification problem would reveal if the overload block type elicits distinguishable EEG patterns. This presents a potential direction for our future research.

### 4.3. MATB-II Subtasks Detection

There have been various approaches in the literature that have addressed the problem of tasks detection from EEG. However, they performed segments classification in multi-class sense, meaning that only one task at a time was present in a segment [40,41,42,43]. On the other hand, our approach better reflects real-life multitasking scenarios where managing multiple tasks within the same time segment is most common. Satisfactory detection results support the hypothesis that we can differentiate between different mental activities (through different subtasks) in multitasking from EEG. We speculate that mental processes related to MATB-II subtasks may activate different neural pathways and our model demonstrated the ability to learn and recognize the respective EEG patterns. This holds particular significance in the context of BCIs, where decoding neural signals into actionable commands for external devices relies on recognition of specific mental processes at play. The experimental design minimized physical movements, ensuring that the observed EEG patterns indeed originated from brain activity, rather than from undesired artifacts. The relevance of the data and the model is especially supported by the diverse nature of MATB-II subtasks that cover a variety of human mental processes and the widespread use of the task itself.

The results of this part also shed a light on the problem of TL level classification. Namely, it is important to note that, in our experiment design, the difference between the blocks is not only in the TL level, but also in the selection of subtasks that contribute to the TL. Firstly, PW is an idle state and is clearly different from other blocks that require some activity. Secondly, LL is different from ML and HL as it does not include TRCK. On the other hand, ML and HL are the same in terms of the set of subtasks that are present, being different only in the number of occurrences of the subtasks. As we have shown that the model can successfully learn to detect tasks, it is possible that, also in our TL classification problem, it learned to detect TRCK and used that information to distinguish LL from ML and HL segments. This could explain low confusion between LL and ML&HL classes. On the contrary, ML and HL could not be differentiated in that way and hence high level of confusion between them. Unfortunately, since precise information about TRCK activity (the reason we excluded that subtask from MATB-II subtasks detection) is not available, additional research (with modified experiment) is needed to confirm this hypothesis. Moreover, this reasoning questions experiment designs in related research in the field of MWL estimation. Namely, since deep neural networks are usually black box models, meaning that we lack the insight into the features the model learns to do the classification, this possibility of the model learning to differentiate between subtasks in multitasking scenarios is often overlooked and TL levels are being increased by adding new subtasks [28,51,54]. In other words, the model may learn to differentiate between subtasks presence in different levels, instead of learning to differentiate between MWL levels.

This study focused on the detection of observable active responses within the experimental scenario. Notably, both SYSM and RMAN subtasks required intermittent monitoring of the area of the screen for anomaly identification. However, our model was trained to detect only instances where participants actively responded to these anomalies. Future research could expand upon this by exploring methods to detect more subtle monitoring activities also, even in the absence of observable action, like those during periods without anomalies (e.g., using eye tracking technology).

Finally, an additional purpose of this part of the study was to verify that the model can learn complex EEG patterns caused by the engagement with the experimental task, verifying its capability to learn MWL-related patterns from the same task, crucial for TL level classification problem. With the satisfactory result for the overall multi-label binary classification, the hypothesis was substantiated. This proves the adequacy of the model architecture and implies its potential to learn other EEG-based classification problems as well.

## 5. Conclusions

The study offers valuable insights in two different domains: 1. EEG-based assessment of MWL and 2. EEG-based detection of distinct cognitive tasks’ activity. It sheds light on EEG pattern recognition in multitasking scenarios under various task load levels, where the task load is modified by changing the number of occurrences of smaller subtasks. More specifically, we design a CNN model that learns to classify varying MATB-II task load levels and to detect individual MATB-II subtasks. The results suggest important limitations in EEG-based MWL estimation in multitasking scenarios where TL is modified quantitatively by changing the number of occurrences of subtasks handled within a given timespan. On the other hand, they show surprisingly promising potential in the field of EEG-based tasks differentiation.

Finally, considering future directions of the research, we recognize two potential approaches. Firstly, since the design of this experiment was such that the task load was successfully handled by participants in general, a follow-up study may include the assessment of task difficulties that are set above participants’ working capacity and the effect on the elicited EEG patterns. Secondly, some alternative methods to modify task load levels might be worth exploring. Namely, a different experiment could be designed so that the task load levels are adjusted by focusing on the qualitative aspect—manipulating the complexity of a single task—rather than varying the number of occurrences of multiple subtasks, like it was done in this study. Employing the same model architecture in this new potential dataset and assessing task load classification results would give us further insights into the feasibility of EEG-based MWL assessment for different types of mental tasks.

## Figures and Tables

**Figure 1 brainsci-14-00149-f001:**
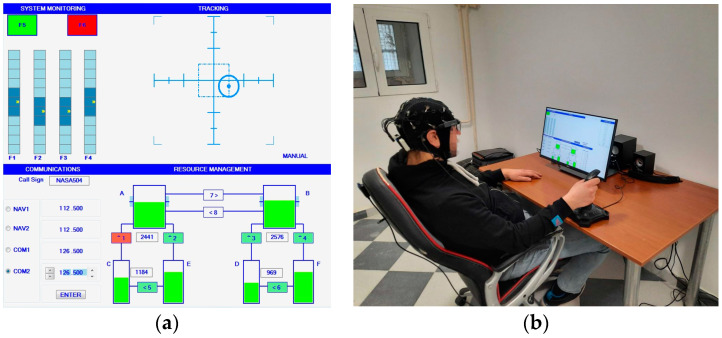
(**a**) MATB-II on the task display; (**b**) Experiment in progress.

**Figure 2 brainsci-14-00149-f002:**
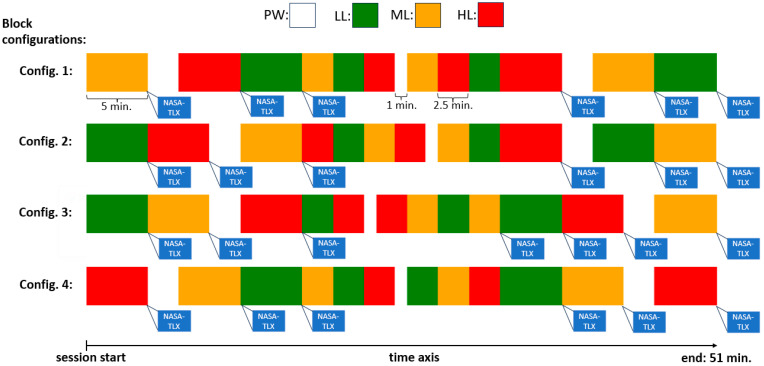
Potential configurations for block arrangement. There were 4 distinct ways each session could be structured. Block types with different task loads are presented with different colors. The instances of administering the NASA-TLX are shown (following each 5 min. block).

**Figure 3 brainsci-14-00149-f003:**
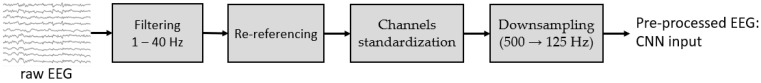
Step-by-step EEG pre-processing.

**Figure 4 brainsci-14-00149-f004:**
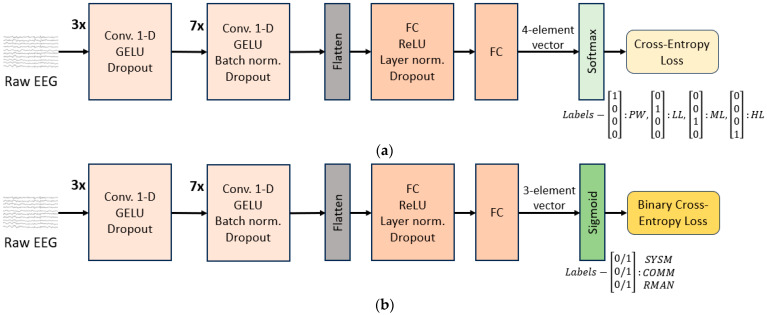
CNN model architecture for (**a**) TL level classification; (**b**) MATB-II subtasks detection. The encoder part consists of 10 1-D 128-channel convolutional layers , each with kernel sizes and strides in the following respective order: kernel sizes (3, 2, 2, 3, 4, 5, 6, 7, 8, 9) and strides (2, 1, 1, 1, 2, 2, 2, 1, 2, 1). Additionally, the second and the third convolutional layers are equipped with skip connections—one over each of the two layers. The model input is of size segment_length_in_seconds × 125 × 25, where 125 is the resampling rate and 25 is the EEG channel number (24 plus one average channel added as part of re-referencing).

**Figure 5 brainsci-14-00149-f005:**
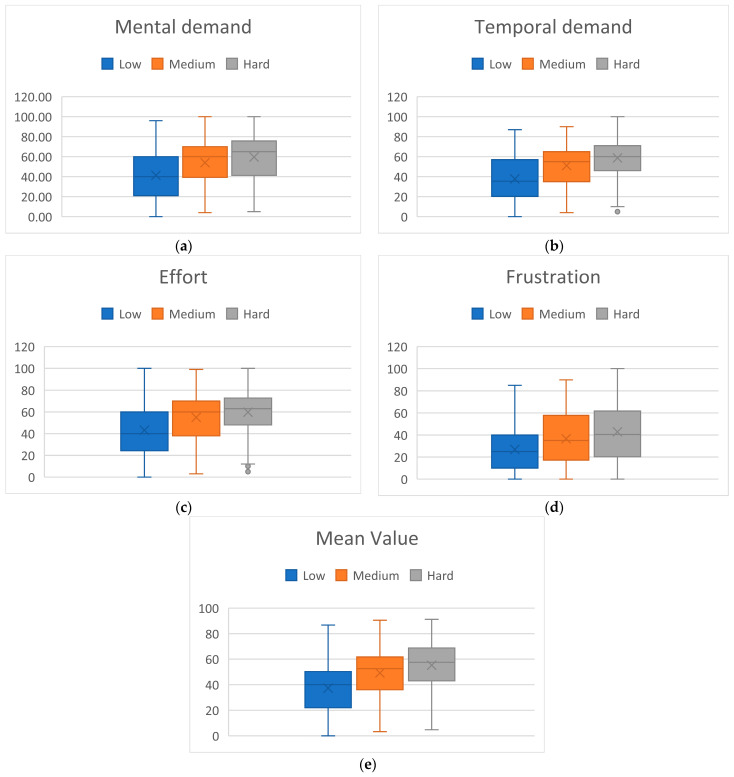
NASA-TLX scores per block type. (**a**) Mental demand: ANOVA *p*-value: 4.98 × $10^{-15}$, Tuckey HSD threshold: 5.23, low vs. medium diff.: 12.62, medium vs. high diff.: 5.60, low vs. high diff.: 18.21; (**b**) Mental demand: ANOVA *p*-value: 4.3 × $10^{-22}$, Tuckey HSD threshold: 4.79, low vs. medium diff.: 13.33, medium vs. high diff.: 7.58, low vs. high diff.: 20.90; (**c**) Effort: ANOVA *p*-value: 8.9 × $10^{-12}$, Tuckey HSD threshold: 5.48, low vs. medium diff.: 11.81, medium vs. high diff.: 4.78, low vs. high diff.: 16.60; (**d**) Frustration: ANOVA *p*-value: 1.2 × $10^{-10}$, Tuckey HSD threshold: 5.46, low vs. medium diff.: 9.66, medium vs. high diff.: 6.30, low vs. high diff.: 15.96; (**e**) Mean value: ANOVA *p*-value: 4.5 × $10^{-18}$, Tuckey HSD threshold: 4.61, low vs. medium diff.: 11.86, medium vs. high diff.: 6.06, low vs. high diff.: 17.92.

**Figure 6 brainsci-14-00149-f006:**
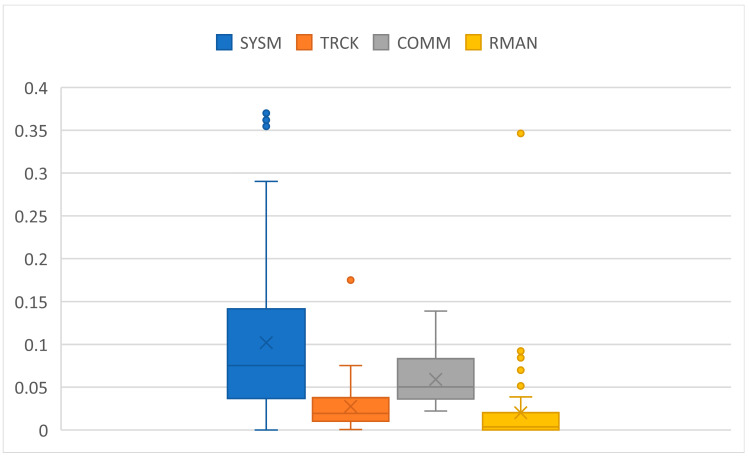
Error rates of subtasks across participants. One point in the graph is the average error rate of the particular subtask and one participant. SYSM: ratio of unsuccessful responses in total number of SYSM calls; TRCK: ratio of time when the circle was out of boundaries (during ML and HL blocks); COMM: ratio of unsuccessful responses in total number of COMM calls; RMAN: ratio of time when tanks A or B liquid levels were out of boundaries (excluding PW blocks).

**Figure 7 brainsci-14-00149-f007:**
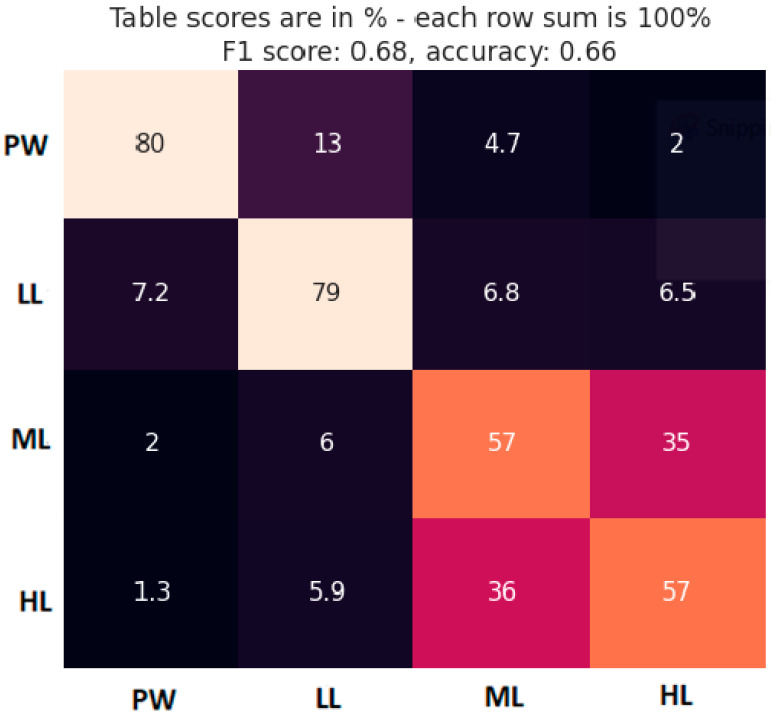
Confusion matrix for block TL classification (true class label in vertical axis). Brighter cell color indicates a higher cell value and vice versa.

**Figure 8 brainsci-14-00149-f008:**
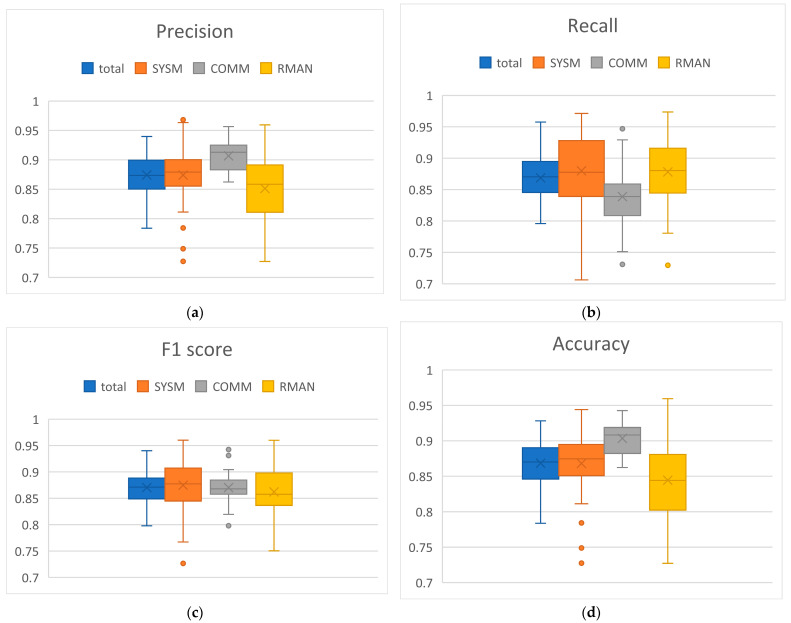
Distributions of subtasks detection results across subjects for different metrics: (**a**) Precision; (**b**) Recall (**c**) F1 score (**d**) Accuracy.

**Figure 9 brainsci-14-00149-f009:**
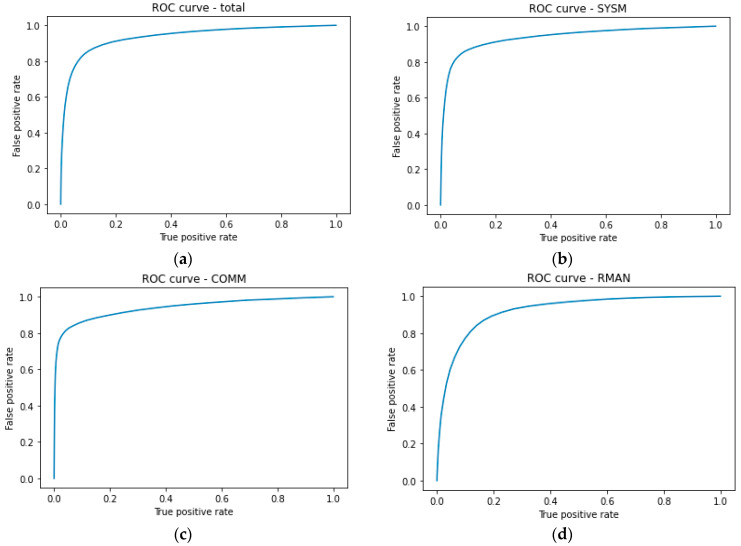
ROC curves for the detection of: (**a**) all subtasks: AUC = 0.94; (**b**) SYSM: AUC = 0.94; (**c**) COMM: AUC = 0.94; (**d**) RMAN: AUC = 0.92.

**Table 1 brainsci-14-00149-t001:** MATB-II subtasks’ occurrences in different block types (PW, LL, ML, HL) in 5 min.

	PW	LL	ML	HL
SYSM	-	10	10	20
TRCK	-	-	Active	Active, faster
COMM	-	6	10	14
RMAN	-	5	10	20

**Table 2 brainsci-14-00149-t002:** F1 score for overall detection and each of the tasks individually.

	Overall	SYSM	COMM	RMAN
F1 score	0.87	0.88	0.87	0.86
Precision	0.87	0.88	0.90	0.85
Recall	0.87	0.88	0.84	0.88
Accuracy	0.87	0.87	0.90	0.84

## Data Availability

The data used in this study were not released publicly due to restrictions imposed by the company involved.

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
