# Peer review of "Mental Workload Classification and Tasks Detection in Multitasking: Deep Learning Insights from EEG Study"

_brainsci, 2024, doi:10.3390/brainsci14020149_

Round 1

Reviewer 1 Report

Comments and Suggestions for Authors

This study explored whether the increased number of subtasks in multitasking led to an increased Mental Workload (MWL) as observed by EEG. Additionally, it examined the detection of activities related to different subtasks from EEG. There are several comments as the following:

1. In the experimental design section, the authors stated that the experiment was divided into 2 sessions (line 209). However, in the subsequent description, it was mentioned that there were 4 sessions (line 231). Please check if there is a contradiction between them.

2. In the MATB-II subtask detection section, the authors mentioned using the method of assigning 0 and 1 to label the presence of subtasks in the segment. Taking the SYSM task as an example, it is a task that requires continuous monitoring. The participants need to allocate their attention resources intermittently to observe if any anomalies occur. If only the occurrence of anomalies and correct responses from the participants are considered as labels indicating the presence of this subtask, then how should the impact of participants' cognitive resources investment when there are no anomalies in the SYSM task be taken into account in terms of cognitive activity and EEG? To address this issue, the authors need to provide more compelling evidence.

3. In the Training Procedure section, further elaboration is needed regarding the division of the training set and the test set. The authors mentioned that "It was done in a way that segments from one session were exclusively in one of the training/validation/test sets", but the experiment only had two sessions. How was this achieved? Additionally, does this statement contradict the claim made in line 375, which states that "5% of the data was used for test and the rest was used for training"?

4. In the experiment results section, line 391 mentions a total of 200 evaluations, but it is not clear how this number was determined. The experiment consisted of four types of blocks: PW, LL, ML, HL. Why the scale analysis only displays three types?

5. The results section could be enriched further, for example, Table 2 could be expanded to include the distribution of results for all participants.

6.  The authors may be interested in adding some discussion with the recent studies on mental workload classification, i.e., Fusion of Spatial, Temporal, and Spectral EEG Signatures Improves Multilevel Cognitive Load Prediction, IEEE Transactions on Human-Machine Systems, 2023.

Comments on the Quality of English Language

The language used in the article is coherent and easy to understand.

Reviewer 2 Report

Comments and Suggestions for Authors

The manuscript, titled "Mental Workload Classification and Tasks Detection in Multi-tasking: Deep Learning Insights from EEG Study," explores a research study utilizing Convolutional Neural Networks (CNN) for EEG pattern recognition in multitasking scenarios across different task load levels. The manipulation of task load involves altering the frequency of smaller subtasks.

The research topic is engaging, and the authors have undertaken an interesting endeavor. Nevertheless, I would like to provide some feedback to the authors:

Introduction:

  1. I kindly recommend the authors consider including references for sentences in lines 73 and 75.
  2. It would be beneficial for the authors to reconsider and enhance the clarity of the sentence in lines 77-79.
  3. In line 81, given the nature of a research paper, I would like to suggest avoiding the use of "common intuition." Additionally, referencing this sentence would be appreciated.
  4. In line 85, rather than stating "these metrics exhibit," it would be preferable to gently emphasize the characteristics of the metrics.
  5. Providing more details about the frequencies extracted in line 91 and considering referencing if possible would greatly enhance the understanding.
  6. Streamlining the information in lines 108-153 would help maintain focus in the introduction.
  7. Highlighting current gaps in the literature, explaining how the authors addressed these gaps, and suggesting future directions would greatly benefit the readers.
  8. The use of bullet points could contribute to improved clarity.
  9. Reorganizing the manuscript for better flow and coherence would be a valuable consideration.
  10. Incorporating a related work section would enhance the overall structure.

Materials and Methods:

  1. It is suggested that a more detailed description of the setup would contribute to a better understanding.
  2. Clarification of line 182 by providing additional information about the protocol and experiment design is kindly recommended.
  3. To avoid redundancy, refraining from defining "Multi-Attribute Task Battery-II (MATB-II) task" again in line 239 would be appreciated.
  4. Providing more information about the subjects, such as their level of instruction, familiarity with BCI, inclusion/exclusion criteria, enrolment, compensation, and recruitment, would be highly valuable.
  5. Including electrode impedance values and specifying if any online filters were used is kindly suggested.

EEG Pre-processing:

  1. Substantially improving this section by providing a step-by-step preprocessing of EEG and audio signals would greatly enhance clarity.
  2. Clearly describing the artifact removal pipeline would be highly appreciated.
  3. Considering the reference to the research paper "Finger pinching and imagination classification: A fusion of CNN architectures for IoMT-enabled BCI applications" could be beneficial.
  4. Addressing the need for a reconstructed 25th epoch and, if found in literature, providing references for clarity would be very helpful.

Deep Learning:

  1. Improving the description of dataset organization, including its size, would contribute to a clearer understanding.
  2. Clearly explaining what was done with the sound signal would be appreciated.
  3. Clarifying the labeling process would contribute to improved comprehension.
  4. Enhancing the explanation of TL extraction and providing a step-by-step description would be highly beneficial.
  5. Adding class labels to Figure 2 for improved clarity is kindly suggested.
  6. Providing more information about each layer's size in the CNN used would enhance understanding.
  7. Offering details on hyperparameters, epochs, learning rate, environment used, libraries, computational cost, time, and resources would be highly informative.
  8. Enhancing Figure 2 by including CNN layer sizes for each block is kindly suggested.
  9. In line 234, providing additional information about the TL extraction process, discussing the length of epochs used, etc., would be appreciated.
  10. Clarifying how TL was extracted for different tasks in each 10s segment and addressing the organization and labeling of data for the CNN would be highly valuable.
  11. Clearly describing the classification approach and comparing results across the SYMM, COMM, and RMAN datasets would contribute to a more comprehensive understanding.

Training Procedure: 

·      I would kindly suggest the authors provide more comprehensive information about the training, test, and validation approach employed, including the percentage of the dataset utilized. 

·      Given the utilization of different datasets and tasks, Figure 3 appears to be somewhat unclear. Addressing this crucial point would significantly enhance the clarity of the methodology. 

·      In Figure 5, the color map is not entirely clear. Additionally, it would be helpful to specify whether the confusion matrix belongs to a specific dataset for better understanding. 

·      For Table 2, I recommend including percentages along with F1, Accuracy, Precision, and Recall to provide a more complete overview. 

·      Considering the use of multiple datasets with multiclassification, incorporating ROC curves could contribute significantly to the research and elevate the manuscript. 

·      It remains unclear whether the authors employed transfer learning, domain adaptation, or trained the CNN from scratch. Additionally, it would be interesting to compare these results with other CNN architectures for a more comprehensive analysis.

Discussion:

I kindly recommend that the authors place greater emphasis on differentiating their work from the current state of the art.

I find this research to be quite intriguing. With these few suggestions, I believe the authors have an opportunity to enhance the overall coherence and impact of their manuscript. Your efforts are appreciated, and I am confident that these refinements will contribute positively to the quality of your work.

Round 2

Reviewer 2 Report

Comments and Suggestions for Authors

The authors did a great job improving the overall manuscript. I would like to provide a further feedback suggesting % where the authors report classification score (e.g 90 %).